# Keratin Dynamics and Spatial Distribution in Wild-Type and K14 R125P Mutant Cells—A Computational Model

**DOI:** 10.3390/ijms21072596

**Published:** 2020-04-09

**Authors:** Marcos Gouveia, Špela Zemljič-Jokhadar, Marko Vidak, Biljana Stojkovič, Jure Derganc, Rui Travasso, Mirjana Liovic

**Affiliations:** 1CFisUC, Center for Physics of the University of Coimbra, Department of Physics, University of Coimbra, R Larga, 3004-516 Coimbra, Portugal; 2Institute for Biophysics, Faculty of Medicine, University of Ljubljana, Vrazov trg 2, 1000 Ljubljana, Slovenia; spela.zemljic-jokhadar@mf.uni-lj.si (Š.Z.-J.); biljana.stojkovic.777@gmail.com (B.S.); jure.derganc@mf.uni-lj.si (J.D.); 3Medical Center for Molecular Biology, Institute for Biochemistry, Faculty of Medicine, University of Ljubljana, Vrazov trg 2, 1000 Ljubljana, Slovenia; marko.vidak@mf.uni-lj.si (M.V.); mirjana.liovic@mf.uni-lj.si (M.L.)

**Keywords:** keratin, epidermolysis bullosa simplex, mutation, phase-field model, reaction-diffusion-advection equation

## Abstract

Keratins are one of the most abundant proteins in epithelial cells. They form a cytoskeletal filament network whose structural organization seriously conditions its function. Dynamic keratin particles and aggregates are often observed at the periphery of mutant keratinocytes related to the hereditary skin disorder epidermolysis bullosa simplex, which is due to mutations in keratins 5 and 14. To account for their emergence in mutant cells, we extended an existing mathematical model of keratin turnover in wild-type cells and developed a novel 2D phase-field model to predict the keratin distribution inside the cell. This model includes the turnover between soluble, particulate and filamentous keratin forms. We assumed that the mutation causes a slowdown in the assembly of an intermediate keratin phase into filaments, and demonstrated that this change is enough to account for the loss of keratin filaments in the cell’s interior and the emergence of keratin particles at its periphery. The developed mathematical model is also particularly tailored to model the spatial distribution of keratins as the cell changes its shape.

## 1. Introduction

The epidermis is the multilayered outer layer of skin, which functions as a protective barrier to all internal tissues and organs. It consists of very tightly packed epithelial cells called keratinocytes. The cytoskeleton of keratinocytes includes keratin intermediate filament (IF) proteins. These are essential to cells since they provide not only mechanical resilience [1,2,3,4], but are also involved in many cell and tissue functions such as cell growth, proliferation, wound healing, migration, etc. [5,6,7,8,9,10,11,12,13,14,15]. Altogether 54 keratin genes have been discovered so far [16]. Apart from the skin, keratins are also abundant in skin appendages such as hair and nails [17,18]. Epidermal keratins are divided into type I and type II proteins. Their genes are clustered on chromosomes 17 (type I) and 12 (type II). Unlike other IF proteins, a type I keratin always pairs up with a specific type II partner, complexifying, thus forming a keratin heterodimer [19,20]. The assembly process of intermediate filaments (IF) is very complex and has been best analyzed for another IF protein, vimentin [21]. Nevertheless the same sequence of events has been observed also for other IF proteins, including keratins: first, the lateral association of rod-like tetrameric complexes of IF proteins results in the so-called ‘‘unit-length filaments’’ (ULFs, 60 nm long structures). ULFs then associate longitudinally (end-to-end) to form shorter filaments, which can subsequently anneal longitudinally to build longer filaments [22].

Mutations affecting keratin genes have been linked to a variety of hereditary cell fragility disorders [23]. The most comprehensibly studied one is epidermolysis bullosa simplex (EBS), a predominantly autosomal dominant disease linked to either keratin 5 or keratin 14 (K5 and K14) gene mutations [24,25]. Their result is the inability of basal layer keratinocytes to resist physical stresses, which manifests as (often severe) skin blistering and wounding. Extensive experiments on EBS patient-derived cell lines have shown that cells retain some of these phenotypic differences also in vitro [26,27,28,29,30,31]. The most typical difference is the presence of highly dynamic keratin particles and aggregates at the cell’s periphery in some keratin mutants [27,28,30,32]. Interestingly, dynamic IF aggregates or even smaller filament fragments have been frequently observed also in normal physiological processes, such as during IF network reorganization [26,27,28,33,34,35,36,37,38,39]. In this respect p38 MAPK has been found as the major regulator of keratin filament remodeling, as well as keratin aggregate formation and disappearance [36]. It has also been shown that keratin precursors appear at the distal tips of actin stress fibers, then move alongside the stress fibers until they integrate the peripheral keratin filament network [35]. Microtubule-dependent transport and dynamics of IF proteins has also been demonstrated both for vimentin and keratin [38,39]. The general structure of an IF protein consists of an extremely conserved central alpha-helical domain, interrupted by two non-helical linkers, and the head and tail end domains, which vary in length and are less conserved. The majority of mutations lie within the central rod domain, and in particular in two highly conserved sequences at its ends (i.e., helix initiation and termination peptide motifs), which have been recognized as important for the assembly of IF filaments [40,41]. The effect of mutations may vary, interfering both at the structural level (at any stage of the filament assembly process), at the interaction with associated proteins or at protein post-translational modifications [27,42].

Recently the understanding of keratin assembly kinetics, turnover and intracellular transport [43] has advanced significantly. In particular, several parameters that determine keratin dynamics in keratinocytes have been measured [44,45], such as the diffusion constant of keratin monomers and the advection velocity of keratin fibers towards the nucleus. Furthermore, mathematical approaches have been used to estimate the spatial dependence and numerical values of association and dissociation rates of keratin monomers [44]. However, none of these studies have addressed keratin dynamics in cells expressing mutant keratin. In this paper we extend a recent model of keratin turnover [44] by employing a 2D phase-field approach to calculate the stationary distribution of keratin in the cell, which mirrors already previously observed conditions in cells in vitro. In particular, our mathematical model accounts for the appearance of keratin particles and aggregates at the cell’s periphery, and may thus be applied also to the case of keratin turnover in mutant keratin expressing cells.

## 2. Results and Discussion

### 2.1. Turnover of Insoluble Keratin

With the aim of providing some visual background information that has led to the development of our improved keratin dynamics mathematical model, we performed a few in vitro experiments, which portray some of the main findings already published on keratin filament dynamics. Immunofluorescent imaging (Figure 1A) of the keratin cytoskeleton of the isogenic EGFP-K14 WT and EGFP-K14 R125P cell lines [30] depicts the structural differences that are often observed between keratin mutant and wild-type keratinocytes. In the mutant EGFP-K14 R125P cells, a lot of tiny keratin particles (blue arrows) are present at the cell’s periphery even when no stress is applied, which is in striking difference from wild-type cells (Figure 1A). As shown in Appendix A, these keratin particles in mutant cells originate at the cell’s periphery where no keratin filaments are visible, assemble into larger particles, and are then either processed or incorporated into the existing keratin filament network. The entire process is very dynamic and only a few minutes elapse between the formation of the particles at the cell’s periphery and their disappearance/incorporation into filaments.

On the other hand, these particles are only rarely observed in the EGFP-K14 WT cells (Appendix A). This is further backed up by a quantitative measurement of the fluorescent keratin signal profile in the direction from the nucleus to the cell’s periphery (Figure 1B): in EGFP-K14 WT cells the signal decreases from approximately 2000 [a.u.] to 500, whereas in mutant cells it increases from approximately 1000 to 2500. We also show the effect of actin filament knockdown (using Cytochalasin D treatment) on keratin filaments and particles movement (Appendix A). After the treatment, the keratin cytoskeleton concentrates around the nucleus, while the majority of keratin particles at the cell’s periphery disappear and the inward movement of particles comes to a halt almost completely.

### 2.2. Mathematical Model of Keratin Cycling in K14 Wild-Type and Mutant Cells

To gain insight into the stationary distributions of keratin in the K14 WT cells and K14 R125P mutant cells, we present here the results of our mathematical model (see Materials and Methods), specifically taking into account the keratin particles observed in the severe keratin mutant(s), by assuming that the mutation affects an intermediate step in keratin assembly (Figure 1C). Our model yields the time evolution and the stationary distribution of the soluble keratin pool in a cell. We considered that keratin is found in three different forms: soluble, particulate, and insoluble (organized into filaments). In a normal situation (WT keratin), the majority of keratin in cells is in its filamentous (insoluble) form, with only a small part of it being present in the soluble form. In mutant keratin expressing cells, on the other hand, this balance is shifted towards a significant increase in the quantity of the soluble fraction, which also contains smaller aggregates of keratin (particulate form) [46]. The keratin concentrations in these forms are represented by $C_{S}$ (soluble), $C_{P}$ (particulate/aggregate) and $C_{I}$ (filaments). The model also takes into account that the particulate and filamentous forms of keratin can bind to the actin filaments and be transported in the direction of the cell’s nucleus. Furthermore, we also considered that the diffusion constant for the soluble form is two orders of magnitude higher than the diffusion constants of the other two forms.

The reaction-diffusion equations for the three forms were solved within the cytoplasm of a generic cell, defined through a phase-field model [47,48]. Importantly, we consider that the keratin filaments can disassemble into the soluble form and that this soluble keratin can assemble back into the keratin particles as shown in the diagram (Figure 1C). The filamentous keratin is the result of the assembly of this intermediate particulate keratin, a process regulated by the rate $K_{PI}$. The values of the parameters of the model are detailed in the Materials and Methods section.

The choice of such keratin kinetics is supported by the particle dynamics mutant keratinocytes display, which is also visible for the EGFP-K14 R125P mutant cell line (Figure 1A and Appendix A). Namely, keratin particles are able to move in the direction of the cell’s nucleus, as well as clearly assemble into filaments. The fact that they are not able to invade the interior of the cell and are restricted to the vicinity of the membrane indicates that keratin particles can disassemble. In other words, the particulate phase is not an absorbing state of the dynamics. In this way, the assembly and disassembly processes described by our equations represent a minimal description of the processes observed in vitro.

We then focused on the effect of the assembly rate, $K_{PI}$, on the distribution of the particulate and insoluble forms in a cell. In the case of higher values of $K_{PI}$, the particulate form is not able to accumulate in the cell, and the keratin is mainly observed in the other two forms, similarly to what is usually observed in wild-type keratinocytes. On the other hand, for lower values of $K_{PI}$ the concentration of keratin in the particulate form increases. In this way, by lowering the value of the rate constant $K_{PI}$, our model can account for the distribution of the different forms of keratin in the severe K14 R125P mutant cells (KEB7 cell line). In the stationary state of our model and for high values of $K_{PI}$, i.e., in the wild-type, keratin filaments always accumulate close to the cell nucleus since they mostly move along the actin filaments (Figure 2A). Similarly, the soluble keratin distribution follows the same pattern as the insoluble (filamentous) keratin: its concentration is higher closer to the cell’s nucleus (Figure 2B). However, due to the higher diffusion constant, the relative variation of the soluble keratin concentration within the cell is lower than the corresponding variation of the concentration of insoluble keratin.

As the value of $K_{PI}$ becomes smaller, the particulate form of keratin accumulates. Strikingly, this form is only able to accumulate in a ring in the neighborhood of the cellular membrane (Figure 2C) and its concentration will be higher when the values for $K_{PI}$ are lower (Figure 2D). In addition, the value of $K_{PI}$ affects the concentration of insoluble keratin filaments only quantitatively. In fact, even with low $K_{PI}$, the concentration of keratin subunits in the filamentous form is higher close to the cell’s nucleus, and lower at the cell membrane. In Figure 2D we plot the sum of the concentrations of the particulate and insoluble filamentous keratins as a function of the position in the cell, for different values of reaction rate, $K_{PI}$. For large values of $K_{PI}$, this sum follows the pattern observed in the wild-type: there are higher levels of insoluble keratin closer to the nucleus compared to the cell membrane. However, as $K_{PI}$ decreases, the total amount of keratin in the insoluble and particulate forms decreases as well but, strikingly, it also accumulates at the cell’s cortex. Therefore, this simple mathematical model predicts that in mutant cells, where the assembly rate of keratin filaments is bottlenecked by a lower value of $K_{PI}$, there should be an accumulation of the non-soluble forms of keratin at the cell’s cortex, and this is in good agreement with the observations on keratin mutant and wild-type cells (Figure 1B).

### 2.3. Discussion Regarding Modeling Choices

These conclusions are robust to large variations in the value of the diffusion constant of the non-soluble keratin forms. In fact, by varying the diffusion constant of the keratin filaments, $D_{I}$, for over two orders of magnitude we observe that the advection of the filamentous keratin dominates diffusion for the lower values of diffusion constant of the keratin filaments, pushing the filaments into the neighborhood of the nucleus in the wild-type cells (see Figure 3A). Moreover, the lower $D_{I}$ is, the larger will be the accumulation of filamentous keratin forms at the nucleus vicinity.

In the mutant case, for low values of $K_{PI}$, we observe an accumulation of particulate keratin at the cellular membrane for all tested values of the ratio $D_{S}/D_{I}$ that lead to a substantial accumulation of the keratin filaments at the nucleus vicinity in the wild-type cell (Figure 3B). Therefore, whenever the diffusion constant of the soluble keratin is much larger than the diffusion constants of the keratin particles and filaments, the keratin filaments accumulate at the nucleus vicinity in the wild-type cells, and the keratin particles accumulate close to the cell membrane for the keratin mutant.

The conclusions are also robust to different expressions for the function $\gamma_{R}\left(\vec{r}\right)$, a localizing function for the reactions. In Figure 3C we plot the concentration of the filamentous plus particulate keratin concentrations for the wild-type and mutant cell for constant keratin filament disassembly rate, and for when the keratin filament disassembly rate is larger at the cell’s center. We still observe, for both situations, that for high values of $K_{PI}$ the keratin filaments accumulate near the nucleus, while for low values of $K_{PI}$ the insoluble keratin accumulates near the cell membrane.

More complex models could have been constructed, more specifically by distinguishing different types of particles, or different types of keratin filaments. A reasonable first step in complexifying the model would be to divide the population of keratin particles in two types: the particles that can merge into fibers (P1) and the particles that cannot (P2). Therefore, distinguishing these two pools of keratin particles the simulated pathway would be the one described in Figure 3D. Since there is no permanent increase in time of the concentration of keratin particles and since the particles can clearly disassemble back into the soluble phase as they move towards the cell nucleus, in Figure 3D we need to include reactions transforming both keratin particle types into soluble keratin. Importantly, this system would have three more reactions than the one in Figure 1C and five more reactions than the model introduced in [44]. In this more complex system, a reasonable choice to regulate the cell mutation would be to increase the reaction rate from keratin particles P1 to keratin particles P2 (by increasing $K_{P1P2}$, for example). This increase would lead to the accumulation of keratins in the form P2. In the model of Figure 1C we consider just one keratin particle phase with a concentration of particles that corresponds to the sum of all types of keratin particles. Therefore, this concentration would be the sum of the concentrations of the two pools of keratin of the more complex model, i.e., $C_{P}=C_{P1}+C_{P2}$. In this complex model the particles can disassemble by two processes with rates $K_{P1S}C_{P1}$ and $K_{P2S}C_{P2}$. Therefore, if $K_{P1S}\approx K_{P2S}=K_{PS}$, the total rate of particle disassembly rate would be
$$K_{P1S}C_{P1}+K_{P2S}C_{P2}\approx K_{PS}\left(C_{P1}+C_{P2}\right)=K_{PS}C_{P}$$
i.e., equal to the disassembly rate for the model in Figure 1C, and independent of the mutation.

Regarding keratin filament assembly, in the more complex model, the keratin particles can assemble into filaments by a single process with rate $K_{PI}C_{P1}$. Note that in the manuscript we model filament assembly by the rate $K_{PI}C_{P}=K_{PI}\left(C_{P1}+C_{P2}\right)$. For the complex model, in the wild-type case, $C_{P2}$ would be approximately zero (low $K_{P1P2}$), and both filament assembly rates (the one from the Figure 1C model and the one from the complex model) are equal. Strikingly, in the mutated cells, $C_{P2}$ would become much higher than $C_{P1}$ and so the simulated model is able to recover the same order of magnitude for the reaction rate obtained in the complex model by decreasing $K_{PI}$, which is exactly how we model the mutation.

In conclusion, while the model simulated in this work does not describe the distinction between different types of the keratin particles (for which we would require more complex models, as the one exemplified above), it is able to recover the same keratin particle distribution in the mutated cells, since it can reproduce the reaction rates of more complex models between the keratin particles and the other keratin phases.

## 3. Materials and Methods

### 3.1. Cell Lines and Culture Conditions

Real-time imaging experiments were performed on two previously reported isogenic cell lines [30] that were engineered by introducing and stably expressing an extra copy of EGFP labeled K14 WT or K14 R125P construct. In brief, the human K14 wild-type cDNA (NM _000526) was originally cloned into the EGFP-C1 vector (Clontech, Mountain View, CA, USA) and tested by transient transfection of control keratinocyte cells (NEB1). This vector was then used to prepare the mutant construct by introducing the K14 R125P mutation using the QuikChange Site-Directed Mutagenesis Kit (Stratagene, San Diego, CA, USA). To generate the stable cell lines used in this study, the K14 wild-type and mutant cDNAs were cut out of the EGFP-C1 constructs and re-cloned into the pLEGFP-C1 (Clontech, Mountain View, CA, USA) retroviral vector using the HindIII and BamHI restriction sites, and the resulting retroviral wild-type and mutant constructs were used to transfect the NEB1 (control) cell line. After single cell cloning and antibiotic selection with G418, clones were tested by Western blotting for the expression level of the EGFP construct, and the ones that were producing the EGFP K14 construct (WT and mutant) at a 1:1 ratio with the endogenous wild-type K14 were expanded and used. Cells were grown in serum-free EpiLife medium supplemented with EpiLife defined growth supplement and gentamicin/amphotericin (Cascade Biologics, Thermo Fischer Scientific, Waltham, MA, USA), at $37{\;}^{\circ }C$ and 5% CO${}_{2}$.

### 3.2. Real-Time Microscopy

Coverslips with live cells were assembled into a perfusion open-closed chamber, a miniature climate box system suitable for cultivation and live cell imaging of eukaryotic cells (POC chamber, H. Saur, Reutlingen, Germany). The chamber temperature was kept at $37{\;}^{\circ }C$ during imaging by a heater. Images (512×512 pixels) were collected using a 100×1.4 NA oil immersion objective on a Nikon C1 confocal system mounted on an inverted microscope Nikon TE2000 Eclipse equipped with a motorized Z stage. Stacks of optical sections 500nm apart were acquired for each time point. Time points were at a 30s interval during a period of 10min. Image analysis and the resulting movies (Appendix A) were done with NIS-Elements AR software.

### 3.3. Cytohalasin-D Experiments

One day after plating, the initial state and dynamics of the keratin cytoskeleton under normal culture conditions in EGFP-K14 R125P cells ($3\times 10^{4}\;\mathrm{cells}/mL$) was first recorded using the live cell imaging set up described above (Appendix A), after which cells were treated with a 10 μm solution of Cytohalasin D (Sigma-Aldrich, St. Louis, MO, USA) in growth medium, for 25 min, at 37 ${}^{\circ }$C in a humidified atmosphere with 5% (v/v) CO${}_{2}$. After treatment, live cell imaging was performed again to record the effects of actin depolymerization on the keratin cytoskeleton (Appendix A).

### 3.4. Assessment of Keratin Distribution in WT and Mutant Cells

NIS Elements (Nikon) software was used to obtain the maximum intensity projections from the confocal images of cells expressing an extra copy of EGFP labeled K14 WT or K14 R125P construct. Representative intensity profiles spanning from the cell nucleus towards the cell border (N=13 for K14 WT cells and N=18 for K14 R125P mutant cells) were then obtained by FiJi image analysis software [49]. Each profile was first interpolated and then rescaled by Mathematica (WolframResearch), so that the coordinate of the nucleus (obtained by manual selection using Fiji software) was set to 0 and the coordinate of the cell border (obtained by manual selection using Fiji software) was set to 1, and the mean of the rescaled intensity profiles was calculated for each cell type.

### 3.5. Mathematical Model

The mathematical model we implemented to describe keratin dynamics is a 2D extension of the work presented by Portet et al. [44], which involved two keratin phases: the soluble keratin pool and the insoluble keratin filaments. In wild-type keratinocytes the keratin filaments are mostly concentrated at the perinuclear region (see Figure 1A,B) [44]. Our model introduces an additional intermediate phase describing the keratin particles observed close to the cellular membrane of mutant cells (see Figure 1A,B). These keratin particles are highly dynamic and can both integrate into filaments, as well as disassemble to the soluble keratin pool (see Figure 1A and Appendix A). We assumed that the mutation affects an intermediate step in the complex scheme of keratin assembly [43] (see Figure 1C). Since biological systems are inherently complex, when constructing a mathematical model we implement a simplified version of the system that takes into account the main mechanisms that drive its evolution. In the particular case of the dynamics of the keratin mutants, we need to describe not only the soluble keratin and the keratin filaments, as was done before in [44], but also keratin particles. To this effect we added two new reactions to the model in [44], and we control the strength of the mutation by changing the value of the reaction rate $K_{PI}$ (see Figure 1C). In the absence of experimental characterization of the assembly and disassembly rates of keratin oligomers and filaments in the keratin mutants, as we make the model more complex we need to be careful to control the number of parameters in the model, while at the same time being able to describe the spatial distribution of the keratin particles in mutant cells.

We have solved the reaction-diffusion-advection equations associated with the model in Figure 1C inside an irregular domain representing the cell cytoplasm, defined by a phase-field order parameter $\phi (\vec{r},t)$: ϕ≈1 inside the cell’s cytoplasm and ϕ≈0 elsewhere i.e., outside the cell and inside its nucleus. Following [47], the concentration of soluble keratin $C_{S}\left(\vec{r},t\right)$ is given by $C_{S}\left(\vec{r},t\right)=c_{S}\left(\vec{r}\right)\phi \left(\vec{r},t\right)$, where $c_{S}\left(\vec{r},t\right)$ is an auxiliary field that can be nonzero outside the cell’s cytoplasm. The concentrations of the keratin in the soluble, particulate and filamentous form obey the following reaction-diffusion-advection equations (that give the time derivative of concentration of the forms as a function of each keratin’s concentration and their spatial derivatives): $$\frac{\partial (\phi c_{S})}{\partial t}=D_{S}\nabla \cdot \left(\phi \nabla c_{S}\right)+\frac{K_{IS}\gamma_{R}\left(\vec{r}\right)\phi c_{I}}{k_{I}+c_{I}}+K_{PS}\gamma_{R}\left(\vec{r}\right)\phi c_{P}-\frac{K_{SP}\delta \left(\vec{r}\right)\phi c_{S}}{k_{S}+c_{S}}$$
$$\frac{\partial (\phi c_{P})}{\partial t}=D_{P}\nabla \cdot \left(\phi \nabla c_{P}\right)-\nabla \cdot \left(\vec{v}\phi c_{P}\right)-K_{PS}\gamma_{R}\left(\vec{r}\right)\phi c_{P}+\frac{K_{SP}\delta \left(\vec{r}\right)\phi c_{S}}{k_{S}+c_{S}}-K_{PI}\phi c_{P}$$
$$\frac{\partial (\phi c_{I})}{\partial t}=D_{I}\nabla \cdot \left(\phi \nabla c_{I}\right)-\nabla \cdot \left(\vec{v}\phi c_{I}\right)-\frac{K_{IS}\gamma_{R}\left(\vec{r}\right)\phi c_{I}}{k_{I}+c_{I}}+K_{PI}\phi c_{P}$$
where $K_{SP}=9.8/s^{-1}$, $K_{IS}=0.99/s^{-1}$, $k_{S}=570\;\mu M$ and $k_{I}=970\;\mu M$ are constants that characterize the Michaelis–Menten dynamics of assembly of the soluble keratin pool and disassembly of the keratin filaments (see Figure 1C); these values were also used by Portet et al. [44]. In these equations $D_{S}$, $D_{P}$ and $D_{I}$ are the diffusion constants of the three keratin forms, $K_{PI}$ and $K_{PS}$ are the reaction rates associated with the formation of filaments from keratin particles and with the disassembly of keratin particles, respectively. $\vec{v}$ is the advection velocity of the keratin particles and filaments along the actin fibers, while $\delta_{R}\left(\vec{r}\right)$ and $\gamma_{R}\left(\vec{r}\right)$ are functions that modulate the reaction kinetics in different regions of the cell. For the diffusion constants we used $D_{S}=0.88\;\mu m^{2}/s^{-1}$ for the soluble form and $D_{I}=D_{P}=0.0088\;\mu m^{2}/s^{-1}$ for the particulate and filamentous keratins [44]. The value for the diffusion constant $D_{S}$ has been measured with precision and published [45]. In regard to keratin filaments, the advection velocity towards the cell’s center is the main driver behind their dynamics. This velocity has been determined to be in the range of 100 to 600 $nm/s^{-1}$ [33,45]. The value we used in our simulation is $|\vec{v}|=150\;nm/s^{-1}$ which is inside the range of typical velocities. To simplify the analysis, we used the same velocity for keratin particles and filaments. The diffusion constant of these two keratin species needs to be small enough for diffusion not to dominate the advection term. Therefore, in this study, we chose values for $D_{I}$ and $D_{P}$ that are 100 times smaller than $D_{S}$. We explored several different values of $D_{I}$ and $D_{P}$ but did not observe significant qualitative differences in the equilibrium spatial distribution of the three phases (refer to Results section).

We used mass-action kinetics for the disassembly of the keratin particles and for its assembly into keratin filaments. We observed the distribution of the different keratin forms when we varied the rate of filament assembly, given by $K_{PI}$. The value used for keratin particle disassembly rate is $K_{PS}=0.0099/s^{-1}$. The model assumes that keratin assembly occurs mainly at the cell membrane [35,50]. Thus, the assembly rate is proportional to the function $\delta_{R}$($\vec{r}$), which is equal to 1 in the neighborhood of the cell membrane, and 0 elsewhere; at the same time the disassembly rates reach their highest value at the nucleus (the function $\gamma_{R}\left(\vec{r}\right)$ decreases linearly from the nucleus, where it is equal to 1, down to the cell membrane, where it becomes equal to 0). This dependence of the disassembly rate on the distance to the nucleus was found in [44] to fit the distribution of keratin in wild-type cells. However, our conclusions do not change even when we chose a constant value for the disassembly rate (i.e., if $\gamma_{R}\left(\vec{r}\right)$ is constant, see above).

## 4. Conclusions

In this paper we have implemented a mathematical model of the keratin turnover in mutant cells that accounts for the appearance of keratin particles at the cell’s periphery. In particular, we extended the recent model of keratin turnover [44] and employed a 2D phase field approach to calculate the stationary distribution of keratin in the cell. The phase-field description is of particular interest, since it can be coupled to force fields describing cell deformation and cell adhesion to other cells or to ECM fibers [51]. This will be of importance to predict the localization of keratin when the cell is deformed under mechanical stress, which is particularly relevant to the mutant cell. Moreover the model can easily be extended to 3D. In a future work we will be using this model to predict how the keratin distribution can affect cell mechanics in mutated cells. An extension of the phase-field model presented in this work is being developed, in order to account for deformations of the cell membrane. At the same time, experimental work was carried out to study how the K14 R125P mutation affects the cortical rigidity of the cell when compared to WT keratinocytes [52].

The cell line with the K14 R125P mutation was selected as the mutation lies in the helix initiation motif of K14 and represents the most frequently mutated amino acid in K14, with over 65 reported cases and several resulting amino acid changes (R125H, R125C, R125S and R125P). Earlier, Herrmann and colleagues [22] analyzed the impact of the R125H change on in vitro filament polymerization, and surprisingly found that the mutant K14 R125H monomers were still able to form functional keratin IFs in vitro when mixed with wild-type K5. The authors deduced that keratin aggregates visible in mutant keratinocytes might not only be consequence of disrupted filament assembly and that other mechanistic explanations are needed. This more complex interplay between wild-type and mutant keratin will be explored in a future work.

## Figures and Tables

**Figure 1 ijms-21-02596-f001:**
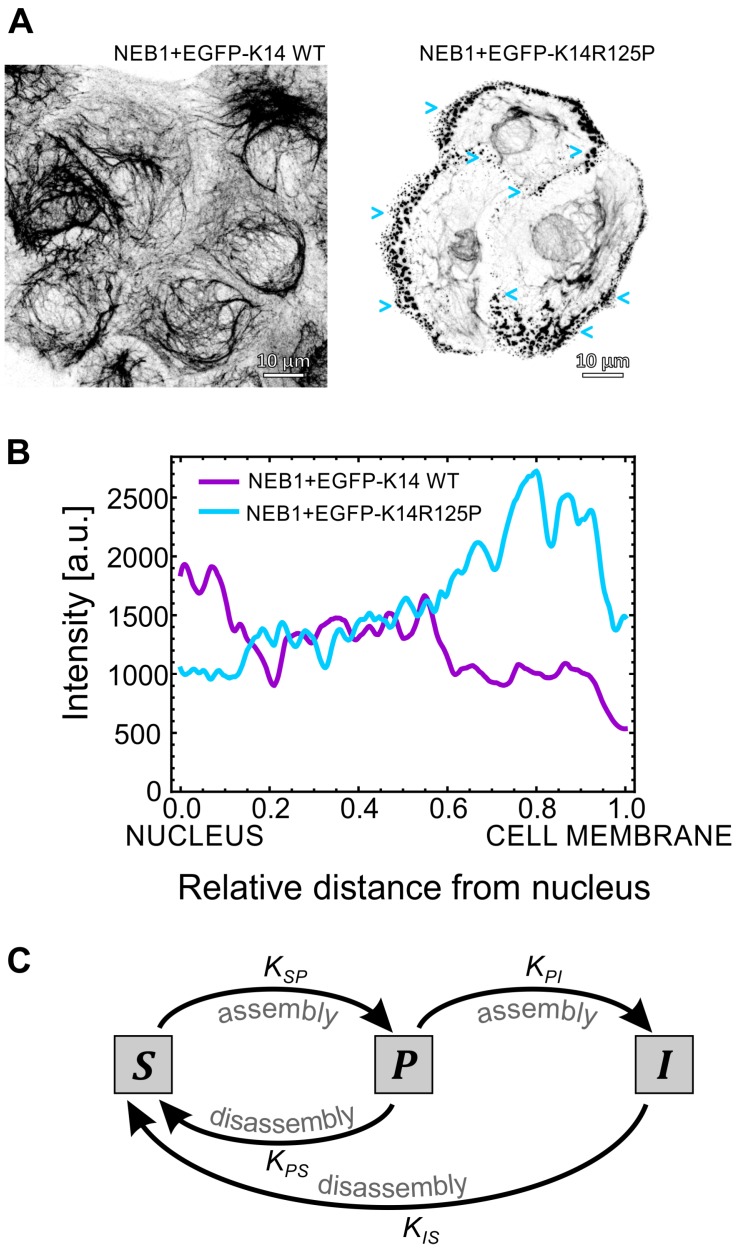
The effect of mutations on keratin turnover. (**A**) Immunofluorescence imaging of the keratin intermediate filament network in the basal cellular part of isogenic (NEB1 background) EGFP-K14 WT and EGFP-K14 R125P mutant keratinocytes (respective movies from which the images were derived from are available as Appendix A). Several small keratin particles (blue arrows) are visible at the cell’s periphery of the K14 R125P mutant. (**B**) The mean measured keratin intensity profiles in WT and mutant cells. (**C**) Diagram representing the kinetic model of keratin assembly. The keratin particles (P) are an intermediate phase in the assembly of the soluble keratin pool (S) into the insoluble keratin filaments (I). The model assumes that the mutation decreases the reaction rate $K_{PI}$ which regulates the assembly of particles into filaments.

**Figure 2 ijms-21-02596-f002:**
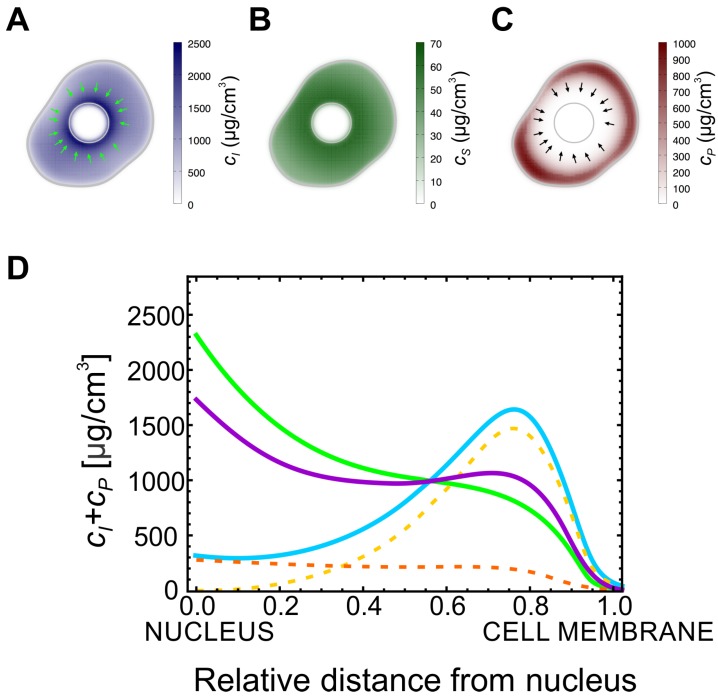
Keratin accumulates at the cell membrane in mutant. (**A**) Typical profile for the insoluble keratin filaments in a cell: the concentration is higher closer to the nucleus. (**B**) Typical profile for the soluble keratin pool concentration: the concentration is also higher closer to the nucleus. (**C**) Typical profile for the concentration of the keratin particles in a mutant cell: the particles are located mainly at the vicinity of the cell membrane. The arrows in (**A**,**C**) point in the direction of the advection velocity of the filaments and particles (respectively). (**D**) Plot of the radially-averaged sum of the concentrations of the particulate and the filamentous keratin forms as a function of the position. The value of $K_{PI}$ increases from the solid blue to the green and to the purple curves ($K_{PI}=0.36,3.6$ and $36\;{\mathrm{hour}}^{-1}$ respectively). In the wild-type most of the non-soluble keratin is located close to the nucleus (purple curve). For lower values of $K_{PI}$ (green and blue curves), the total amount of the non-soluble keratin forms decreases, but the concentration of these forms at the cell cortex is higher. The dashed orange and yellow curves represent the concentration of the keratin filaments and particles, respectively, for the lowest value of $K_{PI}$.

**Figure 3 ijms-21-02596-f003:**
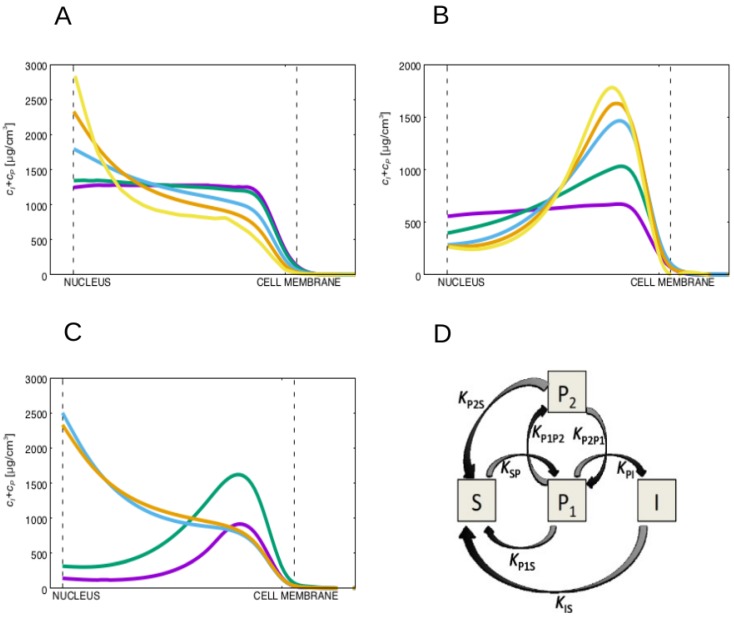
(**A**) Plot of the filamentous plus particulate keratin concentrations for high values of $K_{PI}$ (wild-type) for different diffusion constants, $D_{I}$, of the non-soluble filamentous keratin form. From the purple to the yellow line, $D_{I}$ decreases taking the values $D_{S}$ (purple), $0.1D_{S}$ (teal), $0.02D_{S}$ (blue), $0.01D_{S}$ (orange) and $0.005D_{S}$ (yellow). For all cases $D_{S}=0.88\;\mu m^{2}/s^{-1}$, and the diffusion constant of the particulate keratin is set equal to the diffusion constant of the filamentous keratin. The advection towards the nucleus dominates diffusion for the lower diffusion constants of the keratin filaments, pushing the filaments into the neighborhood of the nucleus. (**B**) Plot of the filamentous plus particulate keratin distributions for low values of $K_{PI}$ (mutant), for different diffusion constants, $D_{I}$, of the filamentous keratin form. From the purple to the yellow line, $D_{I}$ decreases taking the values $D_{S}$ (purple), $0.1D_{S}$ (teal), $0.02D_{S}$ (blue), $0.01D_{S}$ (orange) and $0.005D_{S}$ (yellow). For all cases $D_{S}=0.88\;\mu m^{2}/s^{-1}$, and the diffusion constant for the particulate keratin is set equal to the diffusion constant of the filamentous keratin. We always observe an accumulation of particulate keratin at the cell membrane of the mutant cell for values of $D_{S}\;/\;D_{I}$ large enough to permit the accumulation of the keratin filaments at the nuclear membrane in the wild-type (blue, orange and yellow lines). (**C**) Plot of the filamentous plus particulate keratin concentrations for the high (wild-type, orange and blue curves) and low (mutant, green and purple curves) values of $K_{PI}$. The plots in blue and purple represent the concentrations for constant keratin filament disassembly rate, $K_{SP}$ (in these curves we set $\gamma_{R}\left(\vec{r}\right)=0.5$, i.e., independent of the position within the cytoplasm). The orange and green curves are obtained with the keratin filament disassembly rate larger at the cell center, i.e., they are the same results presented in Figure 2D of the manuscript. We observe that in both situations for high values of $K_{PI}$, the keratin filaments accumulate near the nucleus, while for the low values of $K_{PI}$ the insoluble keratin accumulates near the cell membrane. (**D**) Diagram representing a more complex model for keratin assembly. In this model the keratin particles are separated in two pools P1 and P2, where only keratin particles P1 can polymerize into filaments. We assume in this case that the mutation increases the reaction rate $K_{P1P2}$. This model can describe the distinction between different types of the keratin particles but recovers the same keratin particle spatial distribution as the simpler model.

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
