# Peer review of "Keratin Dynamics and Spatial Distribution in Wild-Type and K14 R125P Mutant Cells—A Computational Model"

_ijms, 2020, doi:10.3390/ijms21072596_

Round 1

Reviewer 1 Report

Sir, 

I have reviewed the manuscript "Keratin Dynamics and Spatial Distribution in Wild-type and Mutant Cells – A Computational Model"  submitted by Marcos Gouveia and co-workers with particular interest. Intermediate filament biology is an interesting topic, and numerous topics in this research area remain enigmatic. Therefore mathematical modelling could shed light on such a complex phenomenon as keratin network dynamics.  

Line 15+16: Introduction appears here twice. Please, omit the second one and also change the numbering of consecutive sections. 

Line 21: wound healing is not a cell function (it is a tissue function). Please, modify this section. 

Line 22-23: Apart from the skin, keratins are also abundant... In fact, nails and hair are also component of the skin (the integument)  .. please, modify this sentence. 

Line 50-51:  The effect of mutations may vary, interfering both at the structural level .......      I would suggest it would be useful here to offer some review publication focusing on available assays to study the consequences of cytoplasmic intermediate filaments mutations. 

Lines 92-94: Authors introduce here three forms of keratin ( soluble, particulate, and insoluble= filamentous). I would suggest that it should be introduced briefly in Introduction with some background information for naïve readers. 

---

I feel that two important questions were omitted in the discussion, and I would like to address them here and ask the authors for a brief reply in the manuscript. 

First, - the study does not induce any physical stress which should be considered in keratinocyte biology and an important external stimulus. I hope this can be easily commented in a short paragraph(and more extensively studied in the future). 

Second - all the keratin dynamics was studied on a simple cell model, and this background, therefore, influences mathematical modelling. However, my hands-on experience with NEB1 says that it is a bit tricky model. The comparison done by authors seems to be fair (K14WT vs. K14 R125P mutant; the genetic background is otherwise equal). We can also neglect here the fact that the study was done on one cell line only (because the genetic background is rather unique and Epidermolysis bullosa is a rare disease!!).  But can we really extrapolate this K14-centric view on all other keratins? Keratin 19 appears under various conditions (causing cell stress) in the skin and oral mucosa and it is a rather unpredictable intermediate filament belonging the same family. I guess that a brief paragraph in the discussion should address the plausibility of extrapolation of this mathematical model on all other keratins (if possible). 

Anyway, I feel that this manuscript is generally well prepared and has very good potential. My overall evaluation is strongly positive. I suggest acceptation after minor changes. 

Author Response

We thank Reviewer #1 for his/hers comments which we will proceed to answer.

1. Line 15+16: Introduction appears here twice. Please, omit the second one and also change the numbering of consecutive sections.

We have corrected this point and the “Introduction” only appears once

2. Wound healing is not a cell function (it is a tissue function). Please, modify this section.

We have altered this sentence to “but are also involved in many cell and tissue functions such as cell growth, proliferation, wound healing, migration, etc [5–15]." Line 19-20

3. Line 22-23: Apart from the skin, keratins are also abundant... In fact, nails and hair are also component of the skin (the integument)... please, modify this sentence.

We have altered this sentence to “Apart from the skin, keratins are also abundant in skin appendages such as hair and nails [17,18].” Line 21-22

4. Line 50-51:  The effect of mutations may vary, interfering both at the structural level .......      I would suggest it would be useful here to offer some review publication focusing on available assays to study the consequences of cytoplasmic intermediate filaments mutations.

We have included now cited at this point reference 27 (Morley et al 2003) and added the following reference:

Haines RL, Lane EB. Keratins and disease at a glance. J Cell Sci. 2012 Sep 1;125(Pt 17):3923-8.

5. Lines 92-94: Authors introduce here three forms of keratin (soluble, particulate, and insoluble= filamentous). I would suggest that it should be introduced briefly in Introduction with some background information for naïve readers.

We have added the following explanation: “In a normal situation (WT keratin), the majority of keratin in cells is in its filamentous (insoluble) form, with only a small part of it being present in the soluble form. In mutant keratin expressing cells, on the other hand, this balance is shifted towards a significant increase in the quantity of the soluble fraction, which also contains smaller aggregates of keratin (particulate form)” Line 94-99

And we now add the following reference:

“Owens DW1, Wilson NJ, Hill AJ, Rugg EL, Porter RM, Hutcheson AM, Quinlan RA, van Heel D, Parkes M, Jewell DP, Campbell SS, Ghosh S, Satsangi J, Lane EB. Human keratin 8 mutations that disturb filament assembly observed in inflammatory bowel disease patients. J Cell Sci. 2004 Apr 15;117(Pt 10):1989-99.”

6. First, - the study does not induce any physical stress which should be considered in keratinocyte biology and an important external stimulus. I hope this can be easily commented in a short paragraph(and more extensively studied in the future).

The aim of this study was to try explain the appearance of keratin aggregates in mutant keratin expressing cells, using an experimental, in silico system. At present it is difficult to also introduce in it the effect of external physical stimuli on wild type and mutant intermediate filament dynamics, and will be extensively studied elsewhere.

The following text was also added to the manuscript’s Conclusion section: “An extension of the phase-field model presented in this work is being developed, in order to account for deformations of the cell membrane. At the same time, experimental work was carried out to study how the K14 R125A mutation affects the cortical rigidity of the cell when compared to WT keratinocytes.”, along with the following reference:

Cortical stiffness of keratinocytes measured by lateral indentation with optical tweezers;Špela Zemljič Jokhadar, Biljana Stojkovic, Marko Vidak, Mirjana Liović, Marcos Gouveia, Rui D.M. Travasso, Jure Derganc;bioRxiv 2020.03.31.018473; doi: https://doi.org/10.1101/2020.03.31.018473

This new paper is the work of the same authors as the paper under review and it is currently being reviewed.

7. Second - all the keratin dynamics was studied on a simple cell model, and this background, therefore, influences mathematical modelling. However, my hands-on experience with NEB1 says that it is a bit tricky model. The comparison done by authors seems to be fair (K14WT vs. K14 R125P mutant; the genetic background is otherwise equal). We can also neglect here the fact that the study was done on one cell line only (because the genetic background is rather unique and Epidermolysis bullosa is a rare disease!!).  But can we really extrapolate this K14-centric view on all other keratins? Keratin 19 appears under various conditions (causing cell stress) in the skin and oral mucosa and it is a rather unpredictable intermediate filament belonging the same family. I guess that a brief paragraph in the discussion should address the plausibility of extrapolation of this mathematical model on all other keratins (if possible).

This model tackles the question of aggregate formation in mutant keratin expressing cells and it is based on data gathered on different keratins and different studies. Nevertheless, the model mimics rather well the situation found in some cases (severe keratin gene mutations correlated with aggregate formation, as discussed in the document). More detailed models may explain further findings on keratin filament dynamics in cells, and will be the object of future studies. In particular, we plan to complexify the simulated keratin dynamics to take into account the keratin response to the stress exerted on cells (see previous comment). We anticipate that a future model would be parameterized to the relevant phenomena in each keratin dynamics, taking into account that the basic mechanisms would be similar.

Reviewer 2 Report

Manuscript ID: ijms-767068

The manuscript entitled 'Keratin Dynamics and Spatial Distribution in Wild-type and Mutant Cells – A Computational Model' by Marcos Gouveia et al. is an interesting manuscript. The authors developed an improved mathematical model for exploring turnover and distribution of keratin in the cells, together with demonstrating some in vitro experimental data. This kind of studies are valuable for predicting of mechanisms in protein dynamics in vivo. The experimental results and methods for construction of mathematical model are clearly presented, and I could understand easily. The reviewer thinks this MS is worth for publication after some modifications and correcting some errors.

  1. Title: ‘Mutant cells’ is not specific presentation. The authors should include specific information of the mutant cells used in this study. The mutant K14 R125P was used for obtaining experimental data and parameters for computational model.

  1. Line 15: ‘1. Introduction’ is duplicated. ‘Introduction’ appears again in line 16.

  1. Legend for Figure 1, lines 4-6: ‘In comparison to the wild type cells, a lot of small keratin particles (blue arrows) are visible at the cell periphery of the K14 R125P mutant even when no stress is applied to cells.’ This sentence should be removed because almost same sentence exists in main text (Lines 71-73).

  1. Lines 229-231: ‘Each profile was first interpolated and then rescaled byMathematica (WolframResearch), so that the coordinate of the nucleus was set to 0 and the coordinate of the cell border was set to 1, and the mean of the rescaled intensity profiles was calculated for each cell type.’ I could not understand how to determine places as the coordinates of the nucleus and the cell border. Please mention about how to determine clearly.

  1. Please recheck whole manuscript again.

Author Response

We thank Reviewer #2 for his/hers comments which we will proceed to answer.

1. Title: ‘Mutant cells’ is not specific presentation. The authors should include specific information of the mutant cells used in this study. The mutant K14 R125P was used for obtaining experimental data and parameters for computational model

We have altered the title of the manuscript to “Keratin Dynamics and Spatial Distribution in Wild-type and K14 R125P Mutant Cells – A Computational Model”, according to the reviewer’s suggestion.

2. Line 15: ‘1. Introduction’ is duplicated. ‘Introduction’ appears again in line 16.

We have corrected this point and the “introduction” only appears once

3. Legend for Figure 1, lines 4-6: ‘In comparison to the wild type cells, a lot of small keratin particles (blue arrows) are visible at the cell periphery of the K14 R125P mutant even when no stress is applied to cells.’ This sentence should be removed because almost same sentence exists in main text (Lines 71-73).

To be different from the main text, we have simplified the sentence in lines 4-6 of the caption to “Several small keratin particles (blue arrows) are visible at the cell periphery of the K14 R125P mutant.”

4. Lines 229-231: ‘Each profile was first interpolated and then rescaled by Mathematica (WolframResearch), so that the coordinate of the nucleus was set to 0 and the coordinate of the cell border was set to 1, and the mean of the rescaled intensity profiles was calculated for each cell type.’ I could not understand how to determine places as the coordinates of the nucleus and the cell border. Please mention about how to determine clearly.

We rewrote the text as:

‘Each profile was first interpolated and then rescaled by Mathematica (WolframResearch), so that the coordinate of the nucleus (obtained by manual selection using Fiji software) was set to 0 and the coordinate of the cell border (obtained by manual selection using Fiji software) was set to 1 , and the mean of the rescaled intensity profiles was calculated for each cell type.’

5. Please recheck whole manuscript again.

We have checked the document again and some minor corrections have been made (text in orange font).